# Probing the temperature of supported platinum nanoparticles under microwave irradiation by in situ and operando XAFS

Taishi Ano[1], Shuntaro Tsubaki [1,2✉], Anyue Liu[1], Masayuki Matsuhisa[1], Satoshi Fujii [1,3], Ken Motokura [1], Wang-Jae Chun [4] & Yuji Wada[1✉]

Microwave irradiation can cause high local temperatures at supported metal nanoparticles, which can enhance reaction rates. Here we discuss the temperature of platinum nanoparticles on $\gamma$-$Al_2O_3$ and $SiO_2$ supports under microwave irradiation using the Debye–Waller factor obtained from in situ extended X-ray absorption fine structure (EXAFS) measurements. Microwave irradiation exhibits considerably smaller Deby–Waller factors than conventional heating, indicating the high local temperature at the nanoparticles. The difference in the average temperatures between the platinum nanoparticles and the bulk under microwaves reaches 26 K and 132 K for $Pt/Al_2O_3$ and $Pt/SiO_2$, respectively. As a result, $Pt/SiO_2$ exhibits considerably more reaction acceleration for the catalytic dehydrogenation of 2-propanol under microwave irradiation than $Pt/Al_2O_3$. We also find microwaves enhance the reduction of $PtO_x$ nanoparticles by using operando X-ray absorption near edge structure (XANES) spectroscopy. The present results indicate that significant local heating of platinum nanoparticles by microwaves is effective for the acceleration of catalytic reactions.

[1] School of Materials and Chemical Technology, Tokyo Institute of Technology, E4-3, 2-12-1, Ookayama, Meguro, Tokyo 152-8552, Japan. [2] PRESTO, Japan Science and Technology Agency (JST), 4-1-8 Honcho, Kawaguchi, Saitama 332-0012, Japan. [3] Department of Information and Communication Systems Engineering, National Institute of Technology Okinawa College, 905 Henoko, Nago, Okinawa 905-2192, Japan. [4] Graduate School of Arts and Sciences, International Christian University, 3-10-2 Osawa, Mitaka, Tokyo 181-8585, Japan. ✉email: shuntaro.tsubaki@gmail.com; yuji-w@apc.titech.ac.jp

Microwave (MW) heating has attracted significant attention as a novel tool for accelerating chemical reactions up to the industrial scales[1–4]. Large MW heating-mediated reaction accelerations have often been reported in gas-solid catalytic reaction systems[5–9]. In contrast to conventional heating (CH), MW energy can be directly supplied to the solid catalysts, which results in focused heating of the solid catalyst. Further, MWs afford shorter reaction times at lower reaction temperatures and improve energy consumption efficiency.

Metal nanoparticles (NPs) interact strongly with MWs and exhibit enhanced catalytic reactions[10–22]. The selective heating of metal NPs by MWs has been considered as a critical factor for the reaction rate enhancements[5]. Several groups have demonstrated the lowering of the reaction temperatures of the NP-mediated catalysis under MW irradiation. For example, Guler et al. demonstrated that the catalytic dehydrogenation of ammonia by Mo NPs was accelerated by MWs, at a temperature that was 250 K lower than that with CH[19]. Similarly, Jie et al. applied a Fe NPs-based catalyst to the dehydrogenation of hydrocarbon fuel and demonstrated that the reaction could be facilitated by MWs[18,21]. The reaction under MWs prevented the generation of light alkane by-products. The authors suggested that the selective and local heating of the supported Fe NPs accelerates the dehydrogenation reaction at the Fe surface and prevents the unexpected side reaction due to thermal cracking.

The impact of the MW-based selective heating on the catalytic reactions has been extensively studied in Pt NPs systems. In their early works, Zhang et al. reported that $CO_2$ reforming of $CH_4$ progressed under MW heating at a temperature that was 100 K lower than that with CH[12]. Meanwhile, Perry et al. reported only a small enhancement of CO oxidation under MW irradiation using $Pt/Al_2O_3$ and $Pd/Al_2O_3$[10]. They undertook a numerical approach and suggested that the rapid heat dissipation from the Pt NPs to gas-phase did not bring about a temperature gradient between the Pt NPs and the surroundings[23]. Silverwood et al. also reported the lack of acceleration in the CO oxidation when the Europt-1 (6 wt% $Pt/SiO_2$) catalyst was employed[14,15]. While extensive studies have been previously conducted to probe these phenomena, the local temperatures of the NPs under MWs have not been well understood due to their very small sizes.

Gaining insights into the temperature distribution in the reaction system is vital for elucidating the mechanism of the reaction enhancement by MWs[5]. Infrared (IR) thermometers are generally used for measuring the external surface temperature of the catalyst bed in the gas-solid catalytic system. However, the internal temperature often becomes higher than the surface temperature, due to which, the measurement of the surface temperature alone can cause an overestimation of the MW-based reaction enhancements. To prevent the overestimation, thermocouples[10,20] or fiber-optic thermometers[8,9,20,24] are inserted into the center of catalyst beds for measuring its core temperature. In contrast, thermography measures the temperature distribution at the surface of the catalyst bed. Besides, numerical simulation can estimate and visualize the complete temperature distribution of the catalyst bed[8,9,24].

The temperature of the NPs at the active site is indispensable for discussing the effects of the local heating on the catalytic reactions under MWs. Kabb et al. demonstrated the ex situ nano-thermometric analysis of the vicinity of Au NPs, which were locally-heated by MWs[25]. Tsukahara et al. have reported that in situ Raman spectroscopy can estimate the local temperature around Co particles with 0.1–3.0 μm diameters[26]. To achieve a similar objective in nano size, Ano et al. had applied a temperature-dependent luminescent lifetime of Rhodamine B as a molecular temperature probe[27]. However, Raman spectroscopy and molecular probe methods are available only in limited conditions. Raman spectroscopy is limited only to highly

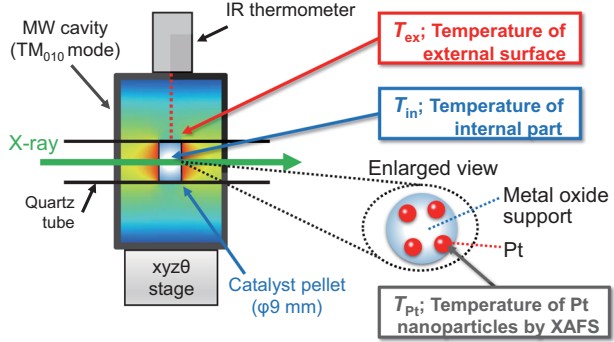

**Fig. 1 Schematic diagram of the in situ XAFS setup under MW heating using $TM_{010}$-mode MW cavity.** $T_{ex}$, $T_{in}$, and $T_{Pt}$ represent the external surface temperature of the catalyst bed, the inner temperature of that, and nanoscale local temperature of Pt NPs, respectively.

Raman-active materials, and the organic molecular probes decompose at the high temperatures. For these reasons, these methods are not suitable for determining local temperatures of metal NPs during gas-solid reactions.

Synchrotron X-ray analysis is a powerful tool for in situ analysis under MW heating[28–34]. In particular, X-ray absorption fine structure (XAFS) spectroscopy affords not only direct structural information of the metal NPs but also the electronic states and the local structures at sub-nanometer scales. Cozzo et al. have demonstrated the first in situ XAFS measurement under MW irradiation for detecting the gelation of Ce solution[31]. Furthermore, Van de Broek et al. have proposed that the Debye–Waller factor ($\sigma^2$) in extended X-ray absorption fine structure (EXAFS) oscillations could be interpreted as the temperate of the Au NPs under Plasmonic heating[35]. Theoretically, the $\sigma^2$ contains thermal disorder ($\sigma_T^2$) and structural disorder ($\sigma_s^2$)[36–39]. In their report, Van de Broek et al. treated the increase of $\sigma^2$ into the $\sigma_T^2$ only because the structure of Au NPs was consistent through the heating, which means that $\sigma^2$ will directly be correlated with the temperature of the Au NPs. These noteworthy results indicate that the local temperature of metal NPs under MW heating can be determined by the in situ XAFS method.

In this work, we demonstrate the nano-thermometric analysis of supported Pt NPs by in situ XAFS spectroscopy under various MW heating conditions. We discuss the local temperature of Pt NPs ($T_{Pt}$, denoted in Fig. 1) from the $\sigma^2$ values. The $T_{Pt}$ exhibits larger temperature than the macroscale temperature distribution in the pellet, indicating the formation of the nanoscale local-high temperature at Pt NPs. We further demonstrates the effect of the local heating on the enhancement of catalytic dehydrogenation of 2-propanol gas. operando X-ray absorption near edge structure (XANES) spectroscopy also confirms the enhanced reduction of the oxidized Pt ($PtO_x$) on $Al_2O_3$ by MW irradiation. Finally, we conclude that the local heating of Pt NPs contributions to the enhanced catalytic reactions by MWs.

## Results and discussion

**Nano-thermometric analysis of Pt NPs by MW in situ EXAFS.** Figure 2a, b shows the TEM image and the XRD patterns of $Pt/Al_2O_3$, respectively. We found that the average particle size was 1.9 ± 0.35 nm referring to the size distribution in Supplementary Fig. 9c. The diffraction peak at $2\theta = 40°$ was assigned to the Pt (111) lattice plane, which indicates that highly dispersed $Pt^0$ NPs were formed on $Al_2O_3$. Figure 2c, d shows in situ FT-EXAFS spectra of $Pt/Al_2O_3$ under CH and MW heating which are

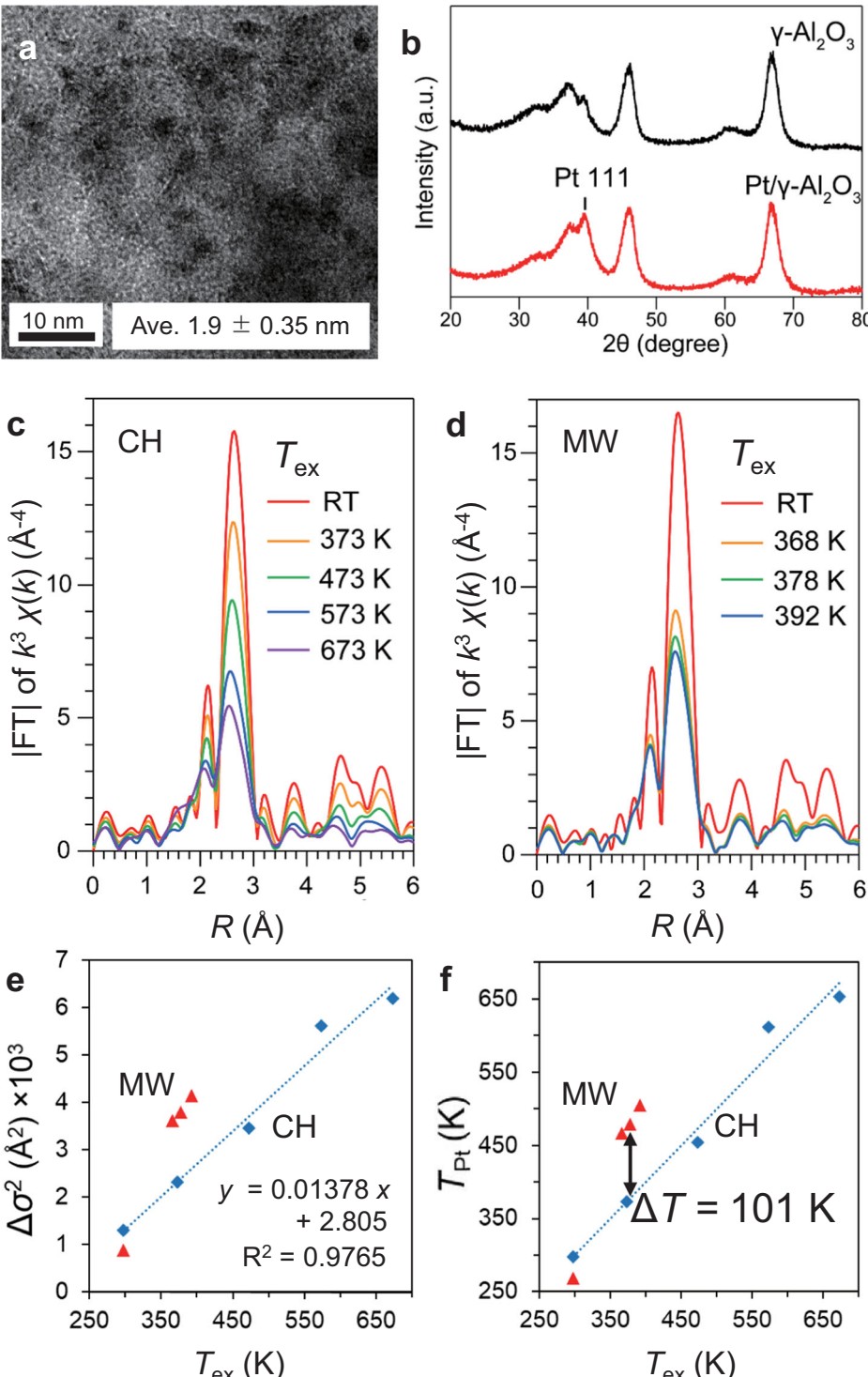

**Fig. 2 In situ EXAFS of Pt/Al₂O₃ under CH and MW heating. a** TEM image and **b** XRD pattern of Pt/Al₂O₃ catalyst. **c** FT-EXAFS spectra obtained by in situ XAFS measurement of Pt/Al₂O₃ catalyst under CH and **d** under MW heating. **e** Relationship between curve-fit $\sigma^2$ and $T_{ex}$. **f** $T_{Pt}$ transformed from $\Delta\sigma^2$ using the linear equation shown in **e**. $\Delta\sigma^2$ values were relative values compared to $\sigma^2$ of Pt foil as reference.

obtained by curve-fitting, respectively (Supplementary Figs. 5–8, Supplementary Tables 2–3). The peaks around 2.7 Å are attributed to the Pt-Pt bonding of Pt NPs. Under CH, the peak intensity gradually decreased as the temperature rises (Fig. 2c). In contrast, the peak intensity under MW heating decreased instantly (Fig. 2d).

The general EXAFS theory (Eq. 1) indicates that the Debye–Waller factor ($\sigma^2$)[40] is an essential factor in the amplitude

dampening of EXAFS oscillation,

$$\chi(k) = S_0^2 \sum_i \frac{N_i F_i(k)}{k R_i^2} e^{-2k^2 \sigma_i^2} \sin[2kR_i + \varphi_i(k)] \quad (1)$$

where $\chi(k)$ is the EXAFS oscillation, $k$ is the wave vector of the excited photoelectron, $N$ is the number of atoms at the interatomic distance ($R$), $F(k)$ is the back-scattering amplitude,

$\varphi(k)$ is the phase shift function. The $\sigma^2$ consists of the thermal vibration factor ($\sigma_T^2$) and the static vibration factor ($\sigma_S^2$), as described in Eq. 2[37,39].

$$\sigma^2 = \sigma_T^2 + \sigma_S^2 \qquad (2)$$

Therefore, it can be assumed that the $\sigma_T^2$ could be isolated if the $\sigma_S^2$ has a constant value, which means no significant structural changes in the Pt NPs through a heating procedure. We found that the TEM results of Pt/Al$_2$O$_3$ after MW heating indicate no changes in the Pt size distribution compared with those before MW heating (Fig. 2a and Supplementary Fig. 9b–d). Further, no significant differences were observed in the curve-fit $\Delta\sigma^2$ before and after heating (Supplementary Fig. 9a). We also found no specific changes in other curve-fit parameters, $N$, $R$, and $\Delta E_0$ under both CH and MW heating (Supplementary Table 3 and Supplementary Fig. 8). These results support that the $\sigma_T^2$ is a crucial factor in the dampening of Pt-Pt peaks in Fig. 2c,d.

Figure 2e shows the plot of the curve-fit $\Delta\sigma^2$ values against $T_{\text{ex}}$. Notably, the $\sigma^2$ values under MW heating increased instantly while the $\Delta\sigma^2$ values under CH increased gradually. Further, a linear relationship was observed between the $\Delta\sigma^2$ and temperature under CH (Fig. 2e). Van de Broek et al. have previously proposed that the $\sigma^2$ values can be used to determine the local temperature of NPs[35]. Therefore, we interpreted the observed changes of the $\Delta\sigma^2$ values obtained under MW heating as those of the local temperature of Pt NPs ($T_{\text{Pt}}$, denoted in Fig. 2f) by applying the linear relationship. The results indicated that the $T_{\text{Pt}}$ was 101 K higher than 378 K, which is the external surface temperature of the pellet measured by the IR thermometer ($T_{\text{ex}}$, Fig. 1).

To discuss the extent of the local heating of the Pt NPs acculately, the $T_{\text{ex}}$ was further corrected to the exact average temperature where the X-ray passes through the internal section of the catalyst pellet ($T_{\text{in}}$, Fig. 1) and the $T_{\text{in}}$ was compared with the $T_{\text{Pt}}$. The temperature distribution of the whole catalyst pellet can be depicted by the coupled simulation[8], toward which we applied COMSOL Multiphysics 5.4a software analysis with electromagnetic field and heat transfer modules. The temperature of the catalyst pellet surface under MW heating (18 W) was first measured by a microscopic thermography with 20 μm resolution (Fig. 3a). The MW setup was exactly reproduced with the one used for the in situ EXAFS (Fig. 3b). The thermographic image indicated the static temperature gradient is formed in the catalyst pellet surface; the center temperature was higher than the edge temperature without spontaneous hot spots. The radial

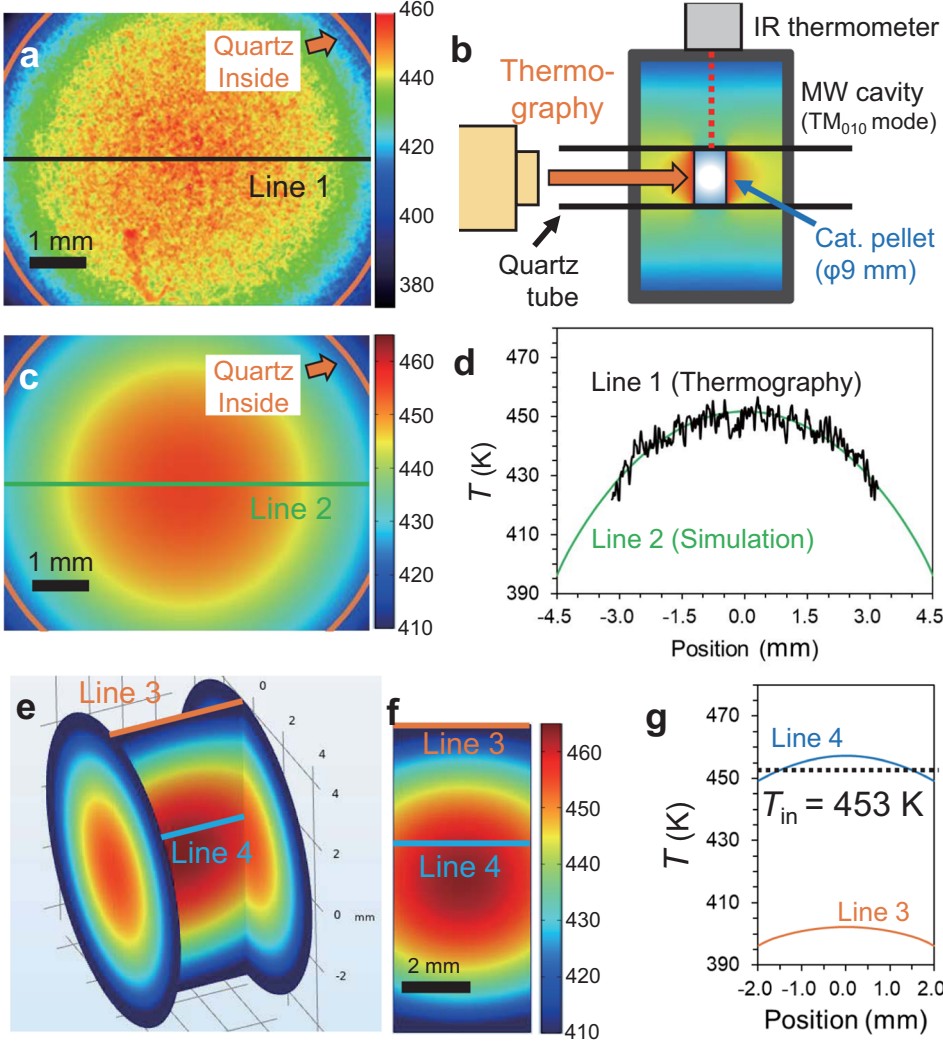

**Fig. 3 Determination of temperature distribution of Pt/Al$_2$O$_3$ catalyst pellet under MW heating. a** Temperature mapping by the thermography under 18-watt MWs. **b** Schematic illustration of MW setup for measuring the temperature mapping of catalyst pellet by the thermography. **c** Reproduced temperature mapping by COMSOL Multiphysics simulation. **d** Temperature profiles of lines 1 and 2. **e** Simulated temperature mapping from a different angle. **f** Temperature profiles of lines 3 and 4. **g** The average temperature of line 4 ($T_{\text{in}}$) along with the X-ray beam during in situ EXAFS experiment.

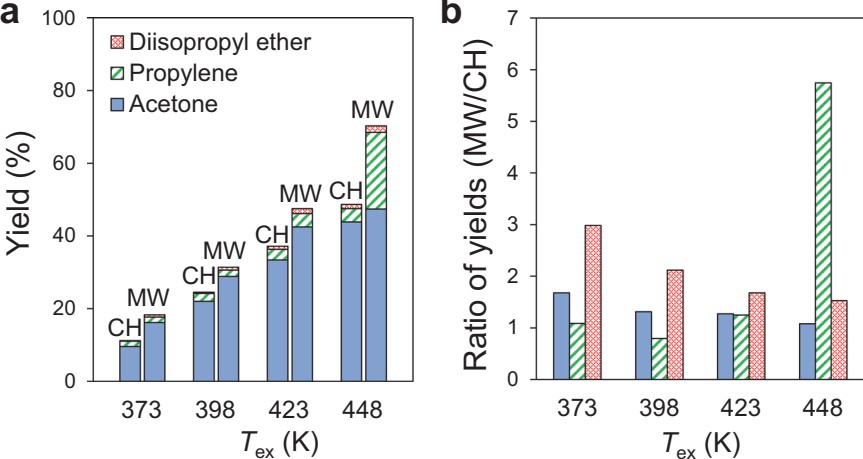

**Fig. 4 Catalytic conversion of 2-propanol by Pt/Al$_2$O$_3$ under MW and CH. a** Product yields by catalytic conversion of 2-propanol with Pt/Al$_2$O$_3$. **b** Comparison of ratios of product yields by MW to those by CH.

temperature gradient was reproduced by the simulation with the effective thermal conductivity of the pellet as $0.29\,\mathrm{W\,m^{-1}\,K^{-1}}$. Figure 3c shows the simulated temperature mapping, and the line profile of the simulation result coincided with that of experimental result (Fig. 3d). The temperature distribution obtained by the simulation for Line 3 axisis is displayed in Fig. 3g. The simulation depicts the whole temperature distribution in the catalyst pellet as shown in Fig. 3e. Line 4 axisis indicates a path in the catalyst bed through which X-ray passes, and the average temperature of the Line 4 ($T_{in}$) was calculated as 453 K. When the $T_{ex}$ was 378 K, the average $T_{in}$ attained 453 K (Supplementary Figs. 15 and 16). Meanwhile, there was no temperature gradient when CH was used (Supplementary Fig. 17). The average $T_{Pt}$ was estimated as 479 K by the in situ XAFS method, and the temperature difference between $T_{Pt}$ and $T_{in}$ was 26 K. This indicates that the Pt NPs supported on Al$_2$O$_3$ are locally-heated under MWs.

Next, we turned to the catalytic conversion of 2-propanol by using the Pt/Al$_2$O$_3$ catalyst under MWs to study the effect of Pt local heating on the catalytic reactions. Figure 4 shows the reaction results under CH and MW heating. Acetone was generated as the main product by the dehydrogenation of 2-propanol at the Pt surface. For instance, the yield of acetone at 373 K was 9.6 % under CH, while the yield by MW attained 16.5 % at the same temperature (Supplementary Fig. 19). This reaction was considerably suppressed without Pt NPs (Supplementary Fig. 20). Since the yield at 398 K under CH was 22.0 %, the reaction enhancement by MWs corresponded to the lowering of the temperature by ~25 K. The production of propylene by the intramolecular dehydration of 2-propanol was also enhanced under MW irradiation. These reaction enhancements should be attributed to the below two aspects of the high-temperature area: The first is the Pt NPs, and the other is the internal part of the catalyst pellet. A fiber-optic thermometer measured the core temperature ($T_{core}$) of the pellet. The $T_{core}$ was higher than the $T_{ex}$ by +8 K (at 373 K) to +13 K (at 448 K) (Supplementary Fig. 22). However, these are too low to explain the above reaction enhancements (~25 K). Therefore, we considered that the reaction enhancement was attributable to both effects of the macroscale local heating and the nanoscale one.

We further applied the experiments to those with Pt/SiO$_2$ catalyst to compare the effect of the metal oxide supports (Fig. 5). Figure 5a shows the average Pt size to be $6.3 \pm 1.8$ nm ($n = 17$), and Fig. 5b shows sharp peaks of Pt (111), Pt (200), and Pt (220) in the XRD pattern. These results indicate that the Pt/SiO$_2$ contains larger Pt$^0$ NPs compared to those in Pt/Al$_2$O$_3$. Further,

the in situ XAFS results of the Pt/SiO$_2$ indicated that the $T_{Pt}$ attained 603 K when the $T_{ex}$ was 376 K (Fig. 5c-f, Supplementary Figs. 10, 11, Supplementary Table 4). The temperature difference was 227 K, which was larger than the Pt/Al$_2$O$_3$. The temperature distribution in the Pt/SiO$_2$ pellet was simulated to obtain $T_{in}$ (Supplementary Fig. 18), which was determined as 471 K when $T_{Pt}$ and $T_{ex}$ were 603 K and 376 K, respectively. Therefore, the temperature difference between $T_{Pt}$ and $T_{in}$ was 132 K.

The yield of the acetone generated using Pt/SiO$_2$ under MWs at 373 K was 2.6 times higher than that with CH (Fig. 6, Supplementary Fig. 21). The yield of acetone generated under MW irradiation at 373 K was 5.9 % between those achieved with CH at 398 K (3.6 %) and 423 K (7.2 %). Thus, the reaction enhancement by MW irradiation corresponds to the decrease in the reaction temperature by ~50 K. Considering the macroscopic temperature distribution, the $T_{core}$ of the Pt/SiO$_2$ catalyst bed was 7 K higher than $T_{ex}$ at 373 K (Supplementary Fig. 22), and thereby, the reaction enhancement by MWs is much larger than that expected by this temperature difference. We conclude that the larger reaction enhancement by Pt/SiO$_2$ is attributable to the nano-sized high temperature of the Pt NPs in Pt/SiO$_2$.

As a result, the local heating realized in the Pt/SiO$_2$ system was more remarkable than that in the Pt/Al$_2$O$_3$ system (Table 1). Accordingly, the Pt/SiO$_2$ catalyst exhibited a larger enhancement of the reaction rate by MWs than that with the Pt/Al$_2$O$_3$ catalyst. Since there were no significant temperature differences between $T_{ex}$ and $T_{core}$ between Pt/Al$_2$O$_3$ and Pt/SiO$_2$, the larger acceleration in Pt/SiO$_2$ catalyst can be explained by the local heating of the Pt NPs. We hypothesize the following four factors that explain the differences in the local heating between the Pt/Al$_2$O$_3$ and the Pt/SiO$_2$.

(1) Heat transfer from the Pt NPs to gas: The size of the Pt NPs (6.3 nm) on the SiO$_2$ support is larger than that of the Pt/Al$_2$O$_3$ (1.9 nm). The surface area of the Pt NPs on SiO$_2$, which are in contact with air, is smaller. Thus, the Pt NPs on SiO$_2$ are not cooled as much as those on Al$_2$O$_3$.

(2) Heat transfer from the Pt NPs to the supports: The thermal conductivity of the SiO$_2$ support was $0.20\,\mathrm{W\,m^{-1}\,K^{-1}}$, which was lower than that of the Al$_2$O$_3$ support ($0.29\,\mathrm{W\,m^{-1}\,K^{-1}}$) (Supplementary Table 7). Moreover, the interfaces of the Pt/supports should be different each other. The surface of $\gamma$-Al$_2$O$_3$ can be defective[41], thereby disallowing a large contact area at the Pt/support interface where heat transfers occur.

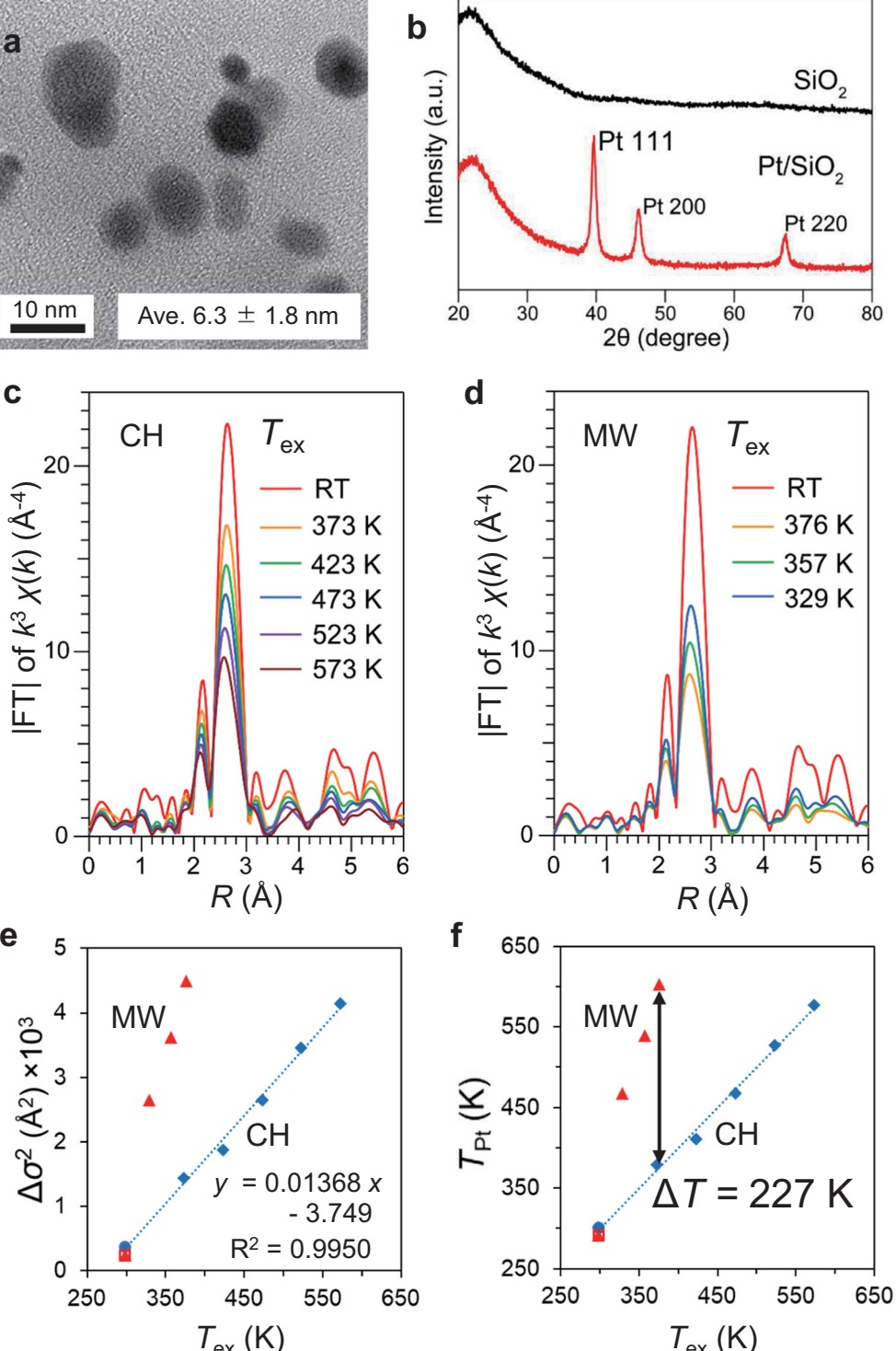

**Fig. 5 In situ EXAFS of Pt/SiO₂ under CH and MW heating. a** TEM image and **b** XRD pattern of Pt/SiO₂ catalyst. **c** FT-EXAFS spectra obtained by in situ XAFS measurement of Pt/SiO₂ catalyst under CH and **d** under MW heating. **e** Relationship between curve-fit $\Delta\sigma^2$ and $T_{ex}$. **f** $T_{Pt}$ transformed from $\Delta\sigma^2$ using the linear equation shown in **e**. $\Delta\sigma^2$ values were relative values compared to $\sigma^2$ of Pt foil as reference.

(3) Selectivity of the Pt heating by MWs: TG results (Supplementary Note 1, Supplementary Figs. 2, 3) showed that the Pt/SiO₂ contains a smaller amount of adsorbed water as compared to the Pt/Al₂O₃. The Pt heating can be more efficient in the Pt/SiO₂ under MWs because there is less microwave absorbers in the Pt/SiO₂ system except for the Pt nanoparticles.

(4) Structural and electric effects of the Pt NPs: Differences in the size, shape, and electrostatically-charging state of the Pt NPs can affect the local MW absorption properties.

**Effect of heat transfer from Pt NPs to gas-phase**. The significant heat transfer to gas-phase has been considered to limit the

occurence of local high temperatures at the supported Pt NPs[23]. The heat dissipation from the particle to gas-phase was predominant because the typical Pt NPs only come into contact with the support in a small area. And then, we have examined the effect of gas flow with different thermal conductivity on the $T_{Pt}$ under MW irradiation.

The in situ XAFS spectra of the Pt/SiO$_2$ pellet under N$_2$ or He flow were measured using the gas flow type system under MWs (Supplementary Fig. 12, Supplementary Table 5). Figure 7 shows that $T_{Pt}$ attained 502 K under N$_2$ flow (10 mL min$^{-1}$) while $T_{ex}$ was 341 K. Thus, their difference reached 161 K. The $T_{Pt}$ under N$_2$ was almost the same as that under air (Fig. 5f). On the other hand, the temperature difference between $T_{Pt}$ and $T_{ex}$ attained only 110 K under He. The less local heating under He can be attributed to the higher thermal conductivity of the He (Supplementary Table 6). Besides, He plasma can also diminish the formation of the local high temperature of the Pt NPs. The He plasma was observed when high MW power, above 60 W, was applied to the catalyst bed (Supplementary Fig. 13). While the plasma was not observed below 50 W, where the in situ XAFS was conducted, the microplasma might be formed and absorb MW energy. This microplasma could contribute to the reduction in the MW energy concentration on the Pt NPs. We consider that the local high-temperature of Pt NPs is affected by the heat dissipation to the gas depending on their thermal conductivity as well as possible formation of microplasma.

**MW operando XANES of the reduction of PtO$_x$/Al$_2$O$_3$.** We conducted operando XANES spectroscopy to explore the local heating effect of PtO$_x$ NPs in the reduction of Pt oxide nanoparticles (PtO$_x$) supported on Al$_2$O$_3$ with 2-propanol. For the reduction process, we supposed that the MW energy absorbed in PtO$_x$ NPs could be efficiently used for the reduction without the heat dissipation to surroundings (namely, 2-propanol and Al$_2$O$_3$). Figure 8a–c shows the normalized XANES spectra of PtO$_x$/Al$_2$O$_3$ under CH. The white line intensities, which appeared at around 11565 eV, gradually decreased over the reaction time at 373 K. Further, the rate of this decrease became larger as the temperature increased. Under MW heating (Fig. 8d, e), notably, the peak intensity decreased drastically even at lower temperatures.

Figure 8g shows the relative rate of PtO$_x$ reduction estimated by the Linear Combination Fitting (LCF) method[42]. The reduction rates under MW heating at 353 K and 373 K were almost identical to those under CH at 423 K and 473 K,

respectively. These results indicate that the MWs promote the PtO$_x$ reduction lowering the reaction temperature by 100 K than that in the CH condition. Also, these operando XANES results are consistent with the in situ EXAFS results that $T_{Pt}$ was 101 K higher than $T_{ex}$, and further indicate that the reaction enhancement is attributed to a thermal effect of the Pt local heating. We conclude that the local heating of Pt NPs is efficient for accelerating the chemical reaction with Pt itself, which constitutes one of the most practical approaches to obtain a dramatic enhancement in catalytic reactions under MW irradiation.

This work demonstrates for the first time, the assessment of the local high temperatures of supported NPs and their effect on catalytic reactions. The nano-thermometric analysis with in situ XAFS is a general method and can be applied to the other metal types of the nanoparticles as long as the structure and the oxidation state of the nanoparticle do not change during heating. As future work, the sizes of the supported Pt NPs should be precisely controlled for a comparative discussion of the support effect on the local heating of the Pt NPs. According to the previous reports, highly endothermic reactions can be more favorable for utilizing the advantages of high local temperatures generated by MWs[5,12]. Moreover, large MW effects have often been observed in catalytic reactions under high-temperature, such as the production of hydrogen from fossil fuels and other materials[21]. Efficient energy concentration at the active sites of the catalysts should be one of the critical strategies for exploring the microwave chemistry to achieve the efficient energy use for reactions and to enable conditions for reaction acceleration at apparent lower temperatures with the use of locally-heated sites.

We apply in situ XAFS spectroscopy under MW heating for the nano-thermometric analysis of supported Pt NPs. The temperature of Pt NPs ($T_{Pt}$) is discussed from the curve-fit $\sigma^2$ values extracted from the structural information measured by in situ EXAFS. The $\sigma^2$ values are further converted to average $T_{Pt}$ using

**Table 1 Summary of temperature differences in $T_{Pt}$, $T_{ex}$, and $T_{in}$.**

| Temperature | Pt/Al$_2$O$_3$ | Pt/SiO$_2$ |
|---|---|---|
| $T_{Pt}$ (K) | 479 | 603 |
| $T_{ex}$ (K) | 378 | 376 |
| $T_{in}$ (K) | 453 | 471 |
| Average MW power (W) | 18 | 58 |

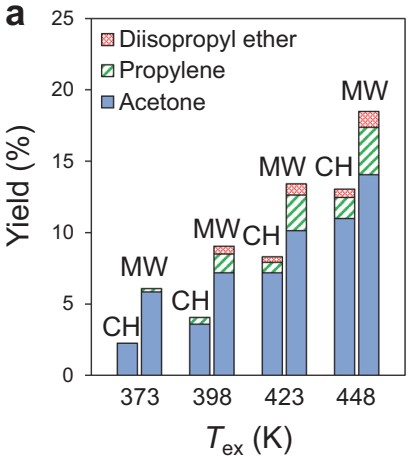
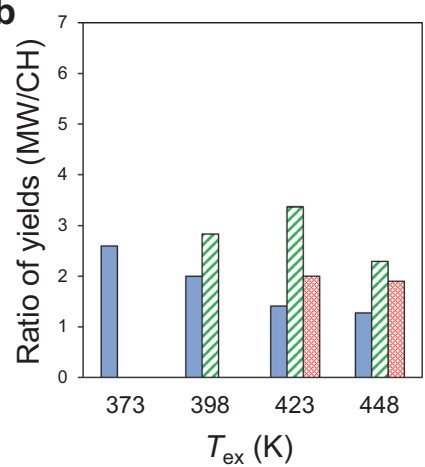

**Fig. 6 Catalytic conversion of 2-propanol by Pt/SiO$_2$ under MW and CH. a** Product yields by catalytic conversion of 2-propanol with Pt/SiO$_2$. **b** Comparison of ratios of product yields by MW to those by CH.

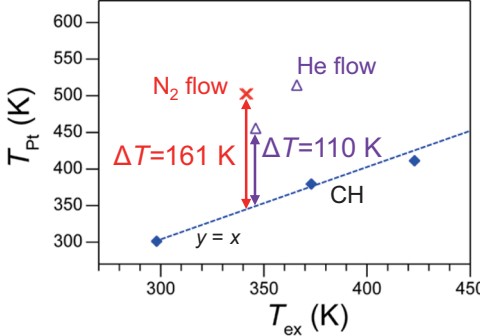

**Fig. 7 $T_{Pt}$ of Pt/SiO$_2$ in N$_2$ or He flow conditions under MWs.** Transformed from $\sigma^2$ using the linear equation shown in Fig. 5e.

the calibration curve obtained by CH. The average $T_{Pt}$ is higher than $T_{ex}$ by 101 K for the Pt/Al$_2$O$_3$ pellet when $T_{ex}$ was 378 K. On the other hand, significant local heating is observed for the Pt/SiO$_2$ system, where the $T_{Pt}$ was 227 K higher than the $T_{ex}$ of 376 K. Furthermore, the dehydrogenation of 2-propanol with Pt/SiO$_2$ proceeds faster compared to that with Pt/Al$_2$O$_3$. Operando XANES further analyzes the extent of the reduction of pre-oxidized PtO$_x$/Al$_2$O$_3$ during dehydrogenation of 2-propanol. The reduction rate of PtO$_x$ under MWs proceeds at temperatures that were 100 K lower to those with the CH condition. The enhancement coincides with the 101 K temperature difference in Pt/Al$_2$O$_3$. Therefore, we conclude that the nano-temperature of Pt under MWs can be determined by the in situ XAFS method, and the Pt local heating has a significant impact on the enhancement of the catalytic reaction rates. These developed methods are examples that enable quantitative verification of the local high temperature formed by MW irradiation. Precise control of the local heating at the supported metal NPs allows the novel design of catalytic reaction processes with the efficient propagation of the MW energy to the active sites.

## Methods

**Catalysts preparation.** Supported Pt NPs catalysts (3 wt%) were prepared by the impregnation method. γ-Al$_2$O$_3$ (97 %, STREM Chemicals Inc.) and SiO$_2$ (99.9 %, Fujifilm Wako Pure Chemicals Co.) were calcined at 973 K for 5 h in the air. The samples were immersed in aqueous H$_2$PtCl$_6$ and dried up at 393 K to obtain the 3 wt% Pt precursors. Pt/Al$_2$O$_3$ and Pt/SiO$_2$ were synthesized by the reduction of the precursors under H$_2$ (36 mL min$^{-1}$) flow at 773 K for 2 h. The calcination of the precursor synthesized PtOx/Al$_2$O$_3$ at 773 K for 5 h in air. Supported Pt NPs were analyzed by FE-TEM (JEM-2010F, JEOL Ltd.) and XRD (MiniFlex600, Rigaku Co. Ltd.). Dielectric properties of as-prepared and vacuum-dried (383 K, 72 h) catalysts were measured by the perturbation method[42,43] using a 2.45 GHz cavity resonator (TM$_{010}$ mode) equipped with a vector network analyzer (ZND, Rohde & Schwarz, Supplementary Table 1). The amounts of water adsorbed on Pt/Al$_2$O$_3$ and Pt/SiO$_2$ were determined using a thermo-gravimetric analyzer (TGA-51, Shimadzu Co.) under 20 mL min$^{-1}$ Ar flow (Supplementary Fig. 1).

**In situ X-ray XAFS under CH and MW heating.** XAFS experiments were conducted in BL 9C beamline at KEK-IMSS-PF (Tsukuba, Japan). Synchrotron radiation from the energy storage ring was monochromatized by Si(111) channel-cut crystals. The monochromator angle was calibrated using Pt foil. Ionization chambers were filled with 15% Ar-85% N$_2$ mixed gas and 100% Ar for monitoring the incident X-ray (I$_0$) and transmitted X-ray (I), respectively. The 11,057–12,662 eV energy range was used for detecting the Pt L3-edge absorption in the QXAFS-mode at 180 s for each scan. Each Pt/Al$_2$O$_3$ and Pt/SiO$_2$ was uniformly mixed with 30 wt% boron nitride (BN) (Fujifilm Wako Pure Chemicals Co.) as a filler and it is pelletized in the quartz tube. The homogeneous pellet is used for following XAFS experiments without unexpected scatterings of the incident X-ray.

The in situ XAFS measurement under CH was applied using an in situ XAFS cell (KEK IMSS PF), which can heat a sample under gas flow (Supplementary Fig. 4a). The sample of 110 mg was pelletized in a metal ring and placed in the in situ XAFS cell. A heat insulator surrounds the in situ XAFS cell to maintain the homogeneous temperature. Two thermocouples measure the temperatures at the sample and flowing gas to keep the same temperature during XAFS measurement. The XAFS spectra were obtained in the temperature range between 298–673 K

XAFS spectra were analyzed with REX2000 software (Rigaku Co., Japan) to obtain the $k^3$-weighted EXAFS spectra and Fourier-transformed EXAFS (FT-EXAFS) spectra using the $k = 3–16$ Å$^{-1}$ range. The Fourier filtering was limited to the $R = 1.9–3.1$ Å range during the curve fitting processes. For estimating precise $\Delta\sigma^2$, curve fit was conducted with empirical $F(k)$ and $\varphi(k)$ extracted from the EXAFS spectra of Pt foil (Nilaco Co.), which were measured by in situ XAFS at 298–673 K under He flow condition to prevent oxidation of Pt.

The in situ XAFS measurement under MW irradiation was conducted using a semiconductor MW generator and the TM$_{010}$-mode cavity resonator (Ryowa Electronics Co., Ltd., Supplementary Fig. 4b). The sample of 180 mg was pelletized in a quartz tube and they were placed in the MW cavity. The temperature was monitored by a quartz-transparent IR thermometer TMSH STM0050 (Japan Sensor Co., Ltd.) to obtain the external surface temperature ($T_{ex}$, Fig. 1) of the pelletized catalyst. MWs were irradiated into the cavity at the constant actual power calculated by subtracting the reflected power of the MW from the incident MW power. Impedance matching was maintained using the slug tuner and frequency-auto-tracking system. The QXAFS spectra were obtained after $T_{ex}$ became constant. The XAFS analysis and curve fitting were conducted by the same method used for the CH conditions indicated above. More details of MW in situ EXAFS was indicated in the Supplementary Note 2.

For the gas flow conditions, the pelletized catalyst sample in a quartz tube was sealed with polyimide (Kapton®) film windows to obtain the MW in situ XAFS spectra under 10 mL min$^{-1}$ N$_2$, or He flow (Fig. 9a). The exact pellet temperature distribution under MW heating ($T_{in}$, Fig. 1) was determined by combining thermography and a coupled simulation of electromagnetic wave and heat transfer[8,9,24].

**Monitoring the temperature distribution of the catalyst.** The temperature distribution in the catalyst pellet under MW heating was determined by a thermography and coupled simulation with electromagnetic field and heat transfer modules. High-resolution thermography (Thermalview X MCR32-XA0350-LWD1.25×, ViewOhre Imaging Co. Ltd., detection wavelength; 7.5–14 μm) was used to obtain the temperature mapping of the surface of the catalyst pellet in the 6.4 × 5.12 mm area with a 20 μm resolution with a temperature range of 293–773 K. The emissivity was determined by calibration with the black body reference (emissivity; 0.94). The sample and the black body (the sample sprayed with the black body) were heated at 423 K on a hotplate. The average temperature of the 9 points of the black body was obtained as about 423 K and obtained emissivity of the Pt/Al$_2$O$_3$ as 0.73 and that of the Pt/SiO$_2$ as 0.78 at 423 K. These values were used for all experiments since there was no substantial difference in the emissivity between 373–473 K.

Coupled simulation analysis was conducted by the finite element method (FEM) using COMSOL Multiphysics 5.4a software (RF and Heat Transfer Modules). Physical properties used in the simulation are summarized in Supplementary Table 7. The simulation model and mesh are shown in Supplementary Fig. 14, which were coupled to electromagnetic waves and heat transfer modules to determine the 3D electric field and temperature distributions in the catalyst pellet. The 2D temperature mapping by thermography was reproduced by the simulation analysis by applying effective thermal conductivity (Supplementary Table 7), and the resulting to obtain the 3D temperature distribution in the catalyst bed. More details of the determination of temperature distribution was indicated in the Supplementary Note 3.

**Catalytic dehydrogenation 2-propanol.** The MW setup is shown in Fig. 9b. Pt/Al$_2$O$_3$ or Pt/SiO$_2$ (0.6–1.0 mm, 200 mg) was packed in a quartz tube reactor (inner diameter of 10 mm). The height of the catalyst bed was 6.0 mm for Pt/Al$_2$O$_3$ and 8.5 mm for Pt/SiO$_2$. 2-Propanol gas (16 mL min$^{-1}$) was introduced into the quartz tube with Ar carrier gas (40 mL min$^{-1}$), and the contact time ($W F^{-1}$) for the 2-propanol gas was 0.21 g h L$^{-1}$. To prevent condensation, the pathway of the 2-propanol gas was heated to 383 K with a ribbon heater. The catalyst bed was heated by CH (electric heating furnace) or MWs. The reaction temperature of the electric furnace (Tokyo Garasu Kikai Co., Ltd.) was controlled with a PID temperature controller equipped with a thermocouple thermometer. For MW heating, another TM$_{010}$-mode cavity with a Q-factor larger than the one of in situ XAFS system was employed. The $T_{ex}$ and the core temperature ($T_{core}$) of catalyst bed were measured with an IR thermometer and fiber-optic thermometer (FSE-35225 Anritsu Meter Co., Ltd.). After pre-heating under Ar flow, 2-propanol was introduced into the reactor using a micro-feeder when the $T_{ex}$ and the $T_{core}$ became constant. The gas and liquid products were collected every 20 min and analyzed by gas chromatography (GC-8A, thermal conductivity detector, Shimadzu Co.) with a Gaskuropack54 column (GL Science Inc.) and GC2014 (Flame ionization detector, Shimadzu Co.) with an Inertcap Pure Wax column (GL Science Inc.), respectively. More details of the catalytic dehydrogenation of 2-propanol was indicated in the Supplementary Note 4.

**Operando XANES of the reduction of PtOx/Al$_2$O$_3$ by 2-propanol.** A pre-oxidized PtO$_x$/Al$_2$O$_3$ sample was measured in the operando XAFS experiment under 2-propanol gas flow to monitor the PtO$_x$ reduction. The operando XANES spectroscopy measurements were conducted by using similar setups applied to

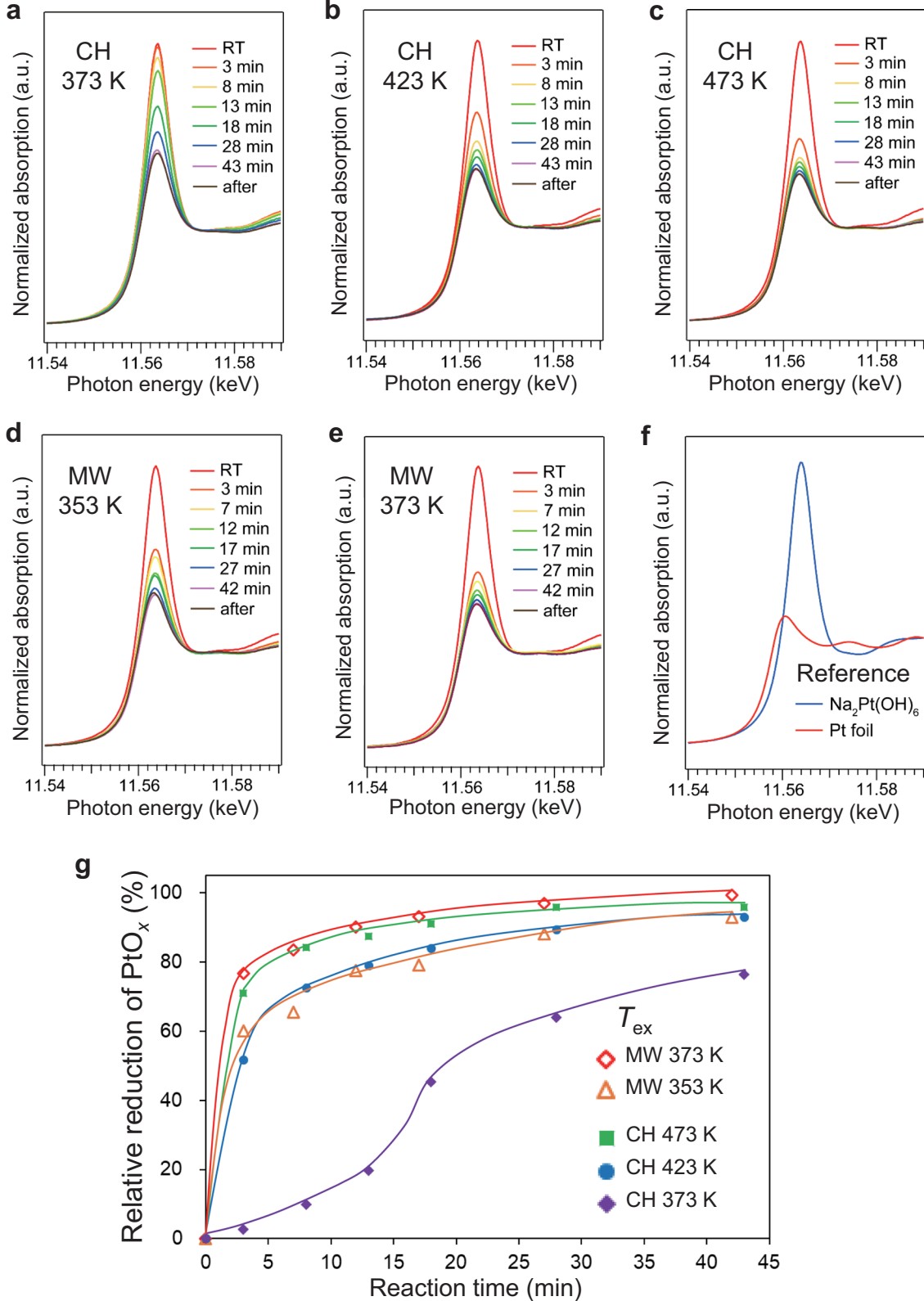

**Fig. 8 Operando XANES of PtO$_x$/Al$_2$O$_3$ during PtO$_x$ reduction reaction by 2-propanol.** Operando XANES spectra obtained under **a–c** CH and **d, e** MWs. **f** References. **g** Relative reduction of PtO$_x$.

in situ XAFS measurement under CH and MW heating, as indicated above (Fig. 9). In the operando measurement under MWs, the catalyst sample was packed in a smaller quartz tube (outer diameter; 9 mm, inner diameter; 7 mm), which was placed inside the larger quartz tube (outer diameter; 12 mm, inner diameter; 10 mm). 2-Propanol gas (5.87 kPa at 298 K) was introduced onto the catalyst bed

with an N$_2$ carrier gas (10 mL min$^{-1}$). QXAFS spectra were obtained every 5 min to trace the progress of the reduction by 2-propanol. According to the previous reports, the Linear Combination Fitting (LCF)[42] method was applied to evaluate the progress of the PtO$_x$ reduction. The decreases of the whiteline intensities were transformed to the relative reductions of PtO$_x$, where the most reduced one

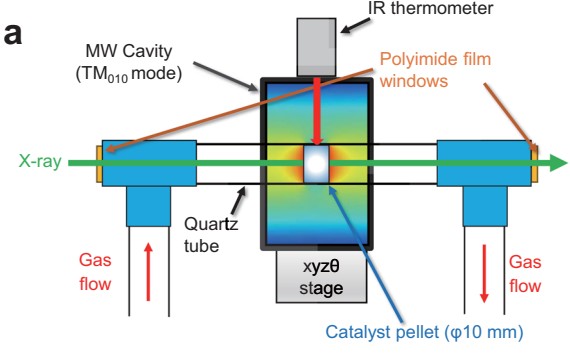

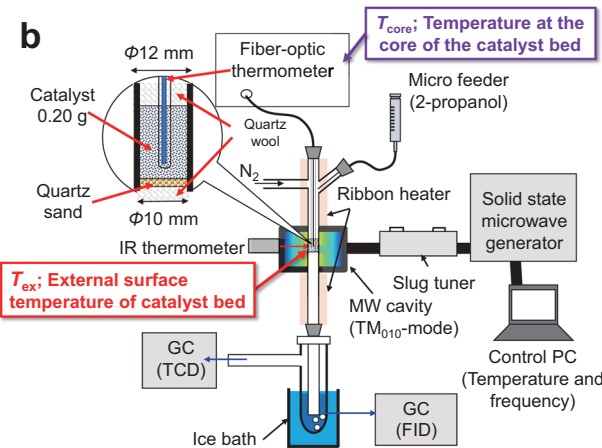

**Fig. 9 Microwave heating setup. a** Schematic diagram of in situ XAFS setup in gas flow condition under MW heating. The catalyst pellet was placed in the center of the $TM_{010}$-mode cavity resonator. The X-ray beam passed through the center of the catalyst pellet. The surface temperature of the catalyst pellet ($Tex$) was monitored by a IR thermometer from the top of the catalyst pellet. **b** Setup for dehydrogenation of 2-propanol by supported Pt NPs under MW irradiation. Tex represents the temperature of the external surface of the catalyst bed measured by the IR thermometer, while the Tcore indicates the core temperature measured by a fiber-optic thermometer.

(MW 373 K for 43 min) was 100%. The XANES spectra of the Pt foil (Nilaco Co.) and Sodium Hexahydroxyplatinate (IV) (Fujifilm Wako Pure Chemicals Co.) were also measured as references.

## Data availability
All the relevant data are indicated in the main text as well as Supplementary Information.

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

## Acknowledgements

TEM measurement was conducted by K. Hori. XAFS measurement was conducted under the approval of the High Energy Accelerator Research Organization, KEK, Photon Factory Advisory Committee (2018G601). This work was supported in part by JSPS Grant-in-Aid for Scientific Research (S) 17H06156; JST PRESTO Grant Number JPMJPR19T6, JSPS Grant-in-Aid for Young Scientists (A) 17H05049 and JSPS Grant-in-Aid Fellows 17J09059.

## Author contributions

T.A. and S.T. contributed to all parts of the work. M.M., K.M. and W.C. contributed to the XAFS experiments. A.L. contributed to the catalytic reactions. S.F. contributed to the simulation work. Y.W. supervised the work.

## Competing interests

The authors declare no competing interests.
