## [Peer Review File · Communications Chemistry]

Reviewers' comments:

Reviewer #1 (Remarks to the Author):

What are the major claims of the paper?

The paper by Tsubaki and coworkers reported a detailed and efficient approach to the determination of real-time internal and external temperature of Pt nanoparticles under MW heating. The impact of silica and alumina supports on the temperatures profiles and gradients is thoroughly treated. The analysis and comparison with CH is well discussed. References are coherent with the data reported and the main issues of the paper are very well introduced. The paper can be accepted for publication after minor revision.

Are they novel and will they be of interest to others in the community and the wider field? YES, strong interest

Is the work convincing, and if not, what further evidence would be required to strengthen the conclusions?

The work is convincing. Nevertheless, the authors should better explain the point 3 on page 16, where a possible influence of the water absorption on NPs surface is indicated without any experimental support. Some data to support this statement need to be reported and discussed. It would be interesting to know if the consideration reported by the authors are of general applicability to other metal NPs beside Pt ones. The authors should indicate possible future improvements of the system (i.e application to other transformations involving gas reagents and/or solvents).

Is there any impact of the reaction scale in the results obtained? Is there any effect of the penetration index of MWs on the results obtained?

The impact of NPs size and support should be properly dissected by starting from NPs with different size on the same solid support? If yes should the author report just a single valuable experiment to indicate the impact of the chemical nature of the support respect to the size?

Is there any effect or observation that should be related to hot spots formation under MW irradiation of the heterogeneous catalysts?

The authors are indicating a possible microwave effect non related only to a difference in temperature observed under MW irradiation with respect with CH. Nevertheless, on page 19 they reported that "the reaction enhancement coincided with the Pt local heating". This issue seems to be disputable and need further explanations.

Reviewer #2 (Remarks to the Author):

In this work the authors present a novel method to determine the nanoparticle temperature under microwave heating by in situ and operando XAFS. According to their calculations, the T_{Pt} of the Pt/Al₂O₃ system under MW heating reached 26 K higher than the bulk Al₂O₃, while in the Pt/SiO₂ system was 132 K higher. This particular topic is of great interest in the field, as knowing the temperature of the catalyst is crucial to determine the effect of MWs in heterogeneous catalysis and explain the observed results that usually imply an enhance of the reaction rates. So far there is no solution to do it accurately, as both optical fibers and IR methods suffer great drawbacks and commonly the rate enhancement can be attributed to simply hot spots. The solution here presented by the authors is well elaborated with an enormous amount of experimental work behind, but some questions need to be resolved prior to consider this temperature measurement technique as valid.

- The external temperature was monitored by a quartz-transparent IR thermometer TMSH STM0050 but no further information is given of this thermometer. What is the wavelength of the thermometer? What is the temperature range? What is the spot size? How do the authors calibrate

the emissivity of the sample? Furthermore, the emissivity varies with temperature, how do the authors consider this fact?

- Regarding the experimental set-up, how is the catalyst bed packed in the tube of Figure 8.b? Is it held by quartz wool?

- Going back to the external temperature, despite the IR thermometer being quartz-transparent, it is located pointing the catalyst bed side in contact with the quartz tube. This implies that the surface measured is obviously colder than the bulk of the bed due to simple heat transfer to the colder and MW transparent quartz tube. This does not take place in conventional heating as the quartz is also heated with the catalyst bed. Therefore, the Text presented by the authors lacks of real meaning and cannot be fairly compared with the T_{Pt} measured by XAFS. At least this fact should be mentioned in the main text.

- As the authors correctly point out, the T_{in} as the bulk temperature should be determined and compared with the T_{Pt} to accurately discuss the extent of the local heating of the Pt NPs. The T_{Pt} measurement seems to be accurate enough with the XAFS and, therefore, measuring T_{in} with sufficient precision is the key here, as it will determine the temperature gradient. This makes section 3 of the SI the most important part of the paper, as the results obtained here depend on the gradient accuracy and the validity of the method. Thus, in my personal opinion, this section should be moved to the main text. Additionally, the experimental part here is not completely clear:

o Again, what is the wavelength of the IR camera? What is the temperature range?

o Where is the pellet located, inside the quartz tube or standing alone in the cavity? Is the MW cavity the same of the XAFS experiment?

o How can the authors discard hot spots in a 5mm thick pellet with just an image of one of the surface sides? Cannot be that the inner pellet is simply hotter due to hot spots and this is the real temperature measured by XAFS and therefore there is no gradient?

o In summary, additional explanations and a scheme, similar to the one in Figure 8, are needed to understand how the measurements were performed.

Reviewer #3 (Remarks to the Author):

Attached review report

Manuscript#: COMMSCHEM-20-0019-T

“Probing nano-temperature of supported Pt nanoparticles under microwave heating by in situ and operando XAFS”

This paper is focused on the measurement of temperature in Pt loaded catalyst under MW heating. This subject is of interest because many of the positive effects claimed for MW heated catalytic reactions, compared to conventional heating, are related to the real temperature measured on the catalyst surface. There is a lot of controversy in this matter and any relevant contribution on temperature measurement is appreciated.

In my opinion the methodology followed in the paper it is not appropriate and the authors are not really measuring “the nano temperature of the supported Pt nanoparticles” as they claim with the XAFS technique that is a bulk characterization technique. Below you will find my comments with this respect and for those reasons the paper should be rejected.

The temperature that they are measuring and correlating with the Debye -Waller factor (σ^2) measured with the X-ray beam (XAFS), is the temperature at the external surface of the pellet, measured with IR thermometer, measured at the outer part of the pellet. In fact, according to their own measurements with the thermographic camera there is a temperature gradient between the external and central part of the pellet and also a temperature gradient within the pellet simulated by COMSOL. At the end in the central part of the pellet, that is where the X-Ray beam (the spot size of the beam should be clearly indicated, it is not mentioned in the paper and SI) is irradiating the sample there is an average temperature of 453K, as it is explicitly presented in the text:

“ T_{in} was then calculated as 453 K, which was the average temperature of the internal section of the catalyst pellet, where the X-ray had passed. When the T_{ex} was 378 K, the simulation revealed T_{in} to be 453 K”

The 453K is the temperature that should be correlated in Figure 2e), showing smaller difference with conventional heating. In fact in the set up for conventional heating, different than the one for MW, it is not clear where they measured the temperature (see below):

“The in-situ XAFS measurement under CH was applied using an in-situ XAFS cell, which can heat a sample under gas flow (Supplementary Fig. 4a). The sample was inserted into a metal ring and was then placed inside the in-situ XAFS cell. The XAFS spectra were obtained in the 298–673 K temperature range, which was controlled by a thermocouple thermometer under air”

Thus, in both cases the authors are measuring similar average bulk temperature in the solid and the temperature difference of 100K that they claim in Figure 2e it is not correct.

Answers to the comments from Reviewer #1:

Comment 1

What are the major claims of the paper?

The paper by Tsubaki and coworkers reported a detailed and efficient approach to the determination of real-time internal and external temperature of Pt nanoparticles under MW heating. The impact of silica and alumina supports on the temperature profiles and gradients is thoroughly treated. The analysis and comparison with CH are well discussed. References are coherent with the data reported and the main issues of the paper are very well introduced. The paper can be accepted for publication after minor revision.

Answer 1

We appreciate the kind comment. We are delighted that the concept of our paper for estimation of local temperature at the Pt nanoparticle under microwaves.

Comment 2

Are they novel and will they be of interest to others in the community and the wider field? YES, strong interest

Answer 2

We appreciate your positive comments, and they encourage us to proceed with this research.

Comment 3

Is the work convincing, and if not, what further evidence would be required to strengthen the conclusions?

The work is convincing. Nevertheless, the authors should better explain the point 3 on page 16, where a possible influence of the water absorption on NPs surface is indicated without any experimental support. Some data to support this statement need to be reported and discussed.

Answer 3

Thank you for the helpful comments. According to the comment, we have revised the point 3 on page 17 as follows:

"Selectivity of the Pt heating by MWs: TG results (Supplementary Fig. 2) showed that the Pt/SiO₂ contains a smaller amount of adsorbed water as compared to the Pt/Al₂O₃. The selectivity of the Pt heating can be higher in the Pt/SiO₂ because there is less microwave absorber in the Pt/SiO₂ system except for the Pt nanoparticles."

The possible contribution of adsorbed water on the catalyst support (especially γ -Al₂O₃) was also discussed in Supplementary Material section 1. Supplementary Fig. 3 indicates

repeated MW heating of the catalyst at different intervals. Adsorbed water induces overheating (Supplementary Fig. 3a) while the 2nd heating immediately after the 1st heating did not (Supplementary Fig. 3b). This is because the desorption of water decreases dielectric loss (Supplementary Table 1). After re-adsorption of water, the overheating again appeared (Supplementary Fig. 3c,d). These heating profiles are strongly dependent on the supports; γ - Al_2O_3 or SiO_2 (Supplementary Table 1). Therefore, the microwave heating profile of the catalysts is also influenced by the adsorption of water molecules on the catalyst.

Supplementary Fig. 3 Temperature profiles during repeated MW heating of Pt/ Al_2O_3 . (a) 1st; Fresh sample, (b) 2nd; sample immediately after the 1st heating, (c) 3rd; sample 1 hour after the 2nd heating, (d) 4th; sample 48 hours after the 3rd heating.

Supplementary Table 1. Dielectric properties of catalysts measured by the perturbation method at 2.45 GHz.

Sample	Before drying			After drying			Weight loss by TG (%)
	ϵ'	ϵ''	$\tan\delta$	ϵ'	ϵ''	$\tan\delta$	
Al ₂ O ₃	2.25	0.180	0.080	1.51	0.029	0.019	-
Pt/Al ₂ O ₃	2.34	0.301	0.128	1.50	0.026	0.017	10.2
SiO ₂	1.22	0.022	0.018	1.11	0.004	0.003	-
Pt/SiO ₂	1.74	0.043	0.025	1.28	0.002	0.002	4.4

Comment 4

It would be interesting to know if the consideration reported by the authors are of general applicability to other metal NPs beside Pt ones. The authors should indicate possible future improvements of the system (i.e application to other transformations involving gas reagents and/or solvents). Is there any impact of the reaction scale in the results obtained? Is there any effect of the penetration index of MWs on the results obtained?

Answer 4

Thank you for the comment. This system applies to various metal-supported catalyst depending on the concentration of supported metal and the interference by the metal oxide support. The gas-solid catalytic system is preferred than the liquid system because the thermal diffusion is smaller in the gas phase and effectively utilize the local high temperature of the nanoparticles. The gas surroundings have another advantage that the gas hardly absorbs microwaves, enabling a large penetration depth of MWs and scale-up of the catalytic reaction system. We have added the following sentences to page 22 in the text.

" In addition, the nano-thermometric analysis with *in-situ* XAFS is a general method and can be applied to the other metal types of the nanoparticles when the structure and the oxidation state of the nanoparticle do not change during heating. "

Comment 5

The impact of NPs size and support should be properly dissect by starting from NPs with different size on the same solid support? If yes should the author report just a single valuable experiment to indicate the impact of the chemical nature of the support respect to the size?

Answer 5

Thank you very much for the comment. We agree with your suggestion. This paper aims to present the novel MW in situ XAFS measurement method for the first observation of the local temperature of supported metal nanoparticles. Our next plan is to determine the local temperature that depends on the metal types, oxidation state, loading amount as well as the size of the particles.

We have added the following discussion to page 22 in the text.

"As future work, the sizes of the supported Pt NPs should be precisely controlled for a comparative discussion of the support effect on the local heating of the Pt NPs. "

Comment 6

Is there any effect or observation that should be related to hot spots formation under MW irradiation of the heterogeneous catalysts?

Answer 6

The thermography with 20 μm resolution showed the static temperature gradient of the catalyst pellet surface, and the inner temperature was higher than the external one, as shown in Supporting Fig. 15a and 17a, but spontaneous hot spots were not detected. We consider that there are temperature gradients along with the radial direction of the pellet (in millimeter scale) as well as at the Pt NPs (in nanoscale).

Comment 7

The authors are indicating a possible microwave effect non related only to a difference in temperature observed under MW irradiation with respect with CH. Nevertheless, on page 19 they reported that "the reaction enhancement coincided with the Pt local heating". This issue seems to be disputable and need further explanations.

Answer 7

We hypothesize two microwave specific effects; one is related to local thermal non-equilibrium, and the other is so-called "non-thermal effects," such as enhancement in the electron transfer process. For instance, our group has previously reported the special effects of MWs on chemical reactions (Wada, Y. et al. Physical insight to microwave special effects: nonequilibrium local heating and acceleration of electron transfer. *J. Jpn. Petrol. Inst.* 61, 98–105 (2018)).

This report focuses on the effect of local high temperature on the gas-solid reaction, and we clearly showed the local temperature of the Pt NPs (T_{Pt}) by *in-situ* EXAFS under MWs, and the Pt temperature was effective for the PtO_x reduction with lowering the temperature by 100 K. This reaction enhancement was consistent with the *in-situ* EXAFS results that T_{Pt} was 101 K higher than the external surface temperature of the catalyst bed measured by the infrared thermometer (T_{ex}). At the moment, we consider that the reaction enhancements by MWs in this report attribute only to a thermal effect of the Pt local heating in this system. After we can completely characterize inhomogeneous temperature distribution in the catalyst bed, we can start discussing the "non-thermal" effect.

Answers to the comments from Reviewer #2:

Comment 1

In this work the authors present a novel method to determine the nanoparticle temperature under microwave heating by *in-situ* and *operando* XAFS. According to their calculations, the TPT of the Pt/Al₂O₃ system under MW heating reached 26 K higher than the bulk Al₂O₃, while in the Pt/SiO₂ system was 132 K higher. This particular topic is of great interest in the field, as knowing the temperature of the catalyst is crucial to determine the effect of MWs in heterogeneous catalysis and explain the observed results that usually imply an enhance of the reaction rates. So far there is no solution to do it accurately, as both optical fibers and IR methods suffer great drawbacks and commonly the rate enhancement can be attributed to simply hot spots.

Answer 1

Thank you for your kind comments. As mentioned by reviewer #2, this method for measuring the local temperature of NPs will be crucial for developing the MWs catalytic system by applying MW selective and local heating. We believe that the NPs temperature will be controlled accurately by designing the structure of NPs on the catalyst support to form a large temperature gradient compared to the surroundings. The local high temperature at the supported NPs can bring about rapid heating of the active site of the catalyst, rapid reaction completion, and high selectivity by reducing the side reaction.

Comment 2

The solution here presented by the authors is well elaborated with an enormous amount of experimental work behind, but some questions need to be resolved prior to consider this temperature measurement technique as valid. The external temperature was monitored by a quartz-transparent IR thermometer TMSH STM0050 but no further information is given of this thermometer. What is the wavelength of the thermometer? What is the temperature range? What is the spot size? How do the authors calibrate the emissivity of the sample? Furthermore, the emissivity varies with temperature, how do the authors consider this fact?

Answer 2

Thank you for your helpful comments. We have added information of IR thermometer (TMSH STM0050), thermography (Thermalview X MCR32-XA0350-LWD1.25x), and the emissivity measurement to the revised manuscript. The IR thermometer with the detection wavelength of 1.95–2.6 μm (temperature range; 50–1000 °C, spot size is φ4 mm) was applied to measure the external temperature of the catalyst bed through the quartz wall in all experiments of *in-situ* EXAFS, catalytic reactions and *operando* XANES.

The thermography with a detection wavelength of 7.5–14 μm (temperature range; 20–500°C, resolution; 20 μm) was used to directly measure the temperature distribution at the surface of the catalyst pellet without quarts.

The emissivity was determined by calibration with the black body reference (emissivity; 0.94). The sample and the black body (the sample sprayed with the black body) were heated at 423 K on a hotplate. The average temperature of the 9 points of the black body was obtained at about 423 K. We determined the emissivity at 423 K of the Pt/Al₂O₃ as 0.73 and that of the Pt/SiO₂ as 0.78. We used these values for calibration of temperature since there was no substantial difference in the emissivity between 373–473 K and will not affect the temperature mapping and the temperature discussion in the catalyst pallet.

Comment 3

Regarding the experimental set-up, how is the catalyst bed packed in the tube of Figure 8.b? Is it hold by quartz wool?

Answer 3

Each Pt/Al₂O₃ and Pt/SiO₂ is mixed with 30 wt% BN filler homogeneously, which is pelletized in the quartz tube without the quartz wool. The mixing with BN enables Pt to be in appropriate the concentration for XAFS measurement. In addition, XAFS measurement needs homogeneity of samples to prevent the unexpected scatterings of the X-ray. We have added the above explanation in the revised manuscript (section for the sample preparation in the *in-situ* EXAFS and *operando* XANES set-up).

Comment 4

Going back to the external temperature, despite the IR thermometer being quartz-transparent, it is located pointing the catalyst bed side in contact with the quartz tube. This implies that the surface measured is obviously colder than the bulk of the bed due to simple heat transfer to the colder and MW transparent quartz tube. This does not take place in conventional heating as the quartz is also heated with the catalyst bed. Therefore, the t_{ext} presented by the authors lack of real meaning and cannot be fairly compared with the T_{Pt} measured by XAFS. At least this fact should be mentioned in the main text.

Answer 4

T_{ex} was measured at the catalyst bed side not to interfere with the XAFS measurement (transmission mode). We agree that the external temperature (T_{ex}) of the catalyst pellet is lower than the internal one (T_{in}) due to the heat transfer to the quartz tube. However, the external temperature is necessary for real-time monitoring of the temperature of the catalyst pellet under MW in situ XAFS measurement, because we cannot measure the core temperature of the pellet during MW in situ XAFS. Therefore, we used simulation to obtain the exact temperature of the internal part of the catalyst

bed (T_{in}). We obtained T_{in} of the catalyst pellet of the *in-situ* EXAFS set up by the thermography and the COMSOL Multiphysics simulation to compare the T_{Pt} and the T_{in} , which are the temperature at the same part of the catalyst pellet where X-ray passed through and evaluated the local temperature of the Pt NPs.

Comment 5

As the authors correctly point out, the T_{in} as the bulk temperature should be determined and compared with the T_{Pt} to accurately discuss the extent of the local heating of the Pt NPs. The T_{Pt} measurement seems to be accurate enough with the XAFS and, therefore, measuring T_{in} with sufficient precision is the key here, as it will determine the temperature gradient. This makes section 3 of the SI the most important part of the paper, as the results obtained here depend on the gradient accuracy and the validity of the method. Thus, in my personal opinion, this section should be moved to the main text. Additionally, the experimental part here is not completely clear: Again, what is the wavelength of the IR camera? What is the temperature range? Where is the pellet located, inside the quartz tube or standing alone in the cavity? Is the MW cavity the same of the XAFS experiment?

Answer 5

Thank you for your helpful comments. We agree with your proposals, and we have transferred this part of the temperature determination of the whole catalyst pellet from Supplementary to the main text in the revised manuscript. Also, the details of the experiments for the 2D temperature mapping by thermography and the simulation for depicting the 3D temperature distribution have been added in the main text. The thermography (Thermalview X MCR32-XA0350-LWD1.25x, ViewOhre Imaging Co. Ltd.) enables the temperature mapping in the 6.4×5.12 mm area with a 20 μ m resolution. The wavelength is 7.5–14 μ m and the temperature range is 293–773 K., The experimental set-up of the thermography, is illustrated in the revised Figure 3b. The MW set-up was exactly reproduced of that used for the *in-situ* XAFS experiment. The catalyst pellet was prepared by pelletizing in the quartz tube and the quartz tube placed at the center of the TM010 cavity resonator.

Comment 6

How can the authors discard hot spots in a 5mm thick pellet with just an image of one of the surface sides? Cannot be that the inner pellet is simply hotter due to hot spots and this is the real temperature measured by XAFS and therefore there is no gradient?

Answer 6

Thank you for the comment. The inhomogeneous distribution of electromagnetic field generally generates hot spots. For instance, the focused electromagnetic field around the bubbles in a solvent (De bruyn et al., J. Am. Chem. Soc. 2017, 139, 15, 5431-5436) or at the contacting points of particles in size of \sim 2 mm (Haneishi et al., Sci. Rep. 2019, 9, 222). In the present experiment, the pellet sample used for MW in situ XAFS

measurement was prepared by fine grinding and palletization to preventing hot spots. This procedure is also important to obtain a good XAFS spectrum. In addition, a solid-state MW generator was used to precisely maintain the applied MW power at the same level to prevent the fluctuation of MW output.

The thermography cannot detect the hot spots in the core catalyst pellet. However, there is no inhomogeneous distribution of the electromagnetic field in the homogeneous catalyst pellet. We were also monitoring in situ resonance frequency during microwave heating but we did not observe any drastic change indicating that there was no sign of formation of an instantaneous hot spot.

Comment 7

In summary, additional explanations and a scheme, similar to the one Figure 8, are needed to understand how the measurements were performed.

Answer 7

Thank you for your helpful comments. As shown in answer 5, the experimental setup with the thermography was drawn in Figure 3b and the details of the temperature mapping results and the methods are added in the revised text.

Answers to the comments from Reviewer #3:

Comment 1

This paper is focused on the measurement of temperature in Pt loaded catalyst under MW heating. This subject is of interest because many of the positive effects claimed for MW heated catalytic reactions, compared to conventional heating, are related to the real temperature measured on the catalyst surface. There is a lot of controversy in this matter and any relevant contribution on temperature measurement is appreciated. In my opinion the methodology followed in the paper it is not appropriate and the authors are not really measuring "the nano temperature of the supported Pt nanoparticles" as they claim with the XAFS technique that is a bulk characterization technique. Below you will find my comments with this respect and for those reasons the paper should be rejected.

Answer 1

Thank you for your helpful comments. As pointed out by reviewer #3, the current transmission XAFS is a bulk analysis with 1 mm × 1 mm of spot size and 4–6 mm thickness of the catalyst pellet. However, XAFS gives information on the average structure in the local region (up to around 6 Å), which means that the present MW in situ XAFS provides average structure local Pt NPs (nano-scale) in a bulk catalyst (millimeter scale).

Besides, similar work has been done by Van de Broek et al., who previously proposed that the Debye-Waller factor (σ^2) in EXAFS oscillations could be applied to estimate the temperate of the Au NPs suspension under Plasmonic heating (Small, 2011, 7, 2498-2506). This paper also indicates the use of the Debye-Waller factor from in situ EXAFS measurement enables estimation of the local temperature of Au NPs.

Another advantage of the current method is high selectivity to the Pt NPs. We used 11057–12662 eV energy of the incident X-ray to exploit only the local information of Pt NPs in the Pt supported catalyst. Conventional thermography is not adequate for material-selective temperature measurement. In addition, nano-scale temperature cannot be measured owing to the diffraction limitation IR. Molecular probes are also not adequate for solid catalysis since they are not stable at high temperature. Therefore, XAFS is one of the few techniques that provide temperature information of NPs under MW irradiation.

Comment 2

The temperature that they are measuring and correlating with the Debye -Waller factor (σ^2) measured with the X-ray beam (XAFS), is the temperature at the external surface of the pellet, measured with IR thermomether, measured at the outer part of the pellet. In fact, according to their own measurements with the thermographic camera there is a temperature gradient between the external and central part of the pellet and also a temperature gradient within the pellet simulated by COMSOL. At the end in the central

part of the pellet, that is where the X-Ray beam (the spot size of the beam should be clearly indicated, it is not mention in the paper and SI) is irradiating the sample there is an average temperature of 453K, as it is explicitly presented in the text: " T_{in} was then calculated as 453 K, which was the average temperature of the internal section of the catalyst pellet, where the X-ray had passed. When the T_{ex} was 378 K, the simulation revealed T_{in} to be 453 K" The 453K is the temperature that should be correlated in Figure 2e), showing smaller difference with conventional heating.

Answer 2

Thank you for the comment. We agree that the T_{in} of the inner part in the catalyst pellet where X-ray passes through should be used to discuss the T_{Pt} of the Pt NPs obtained by *in-situ* XAFS. We indicated all T_{ex} , T_{in} and T_{Pt} values in the paper to describe the overall temperature gradient in the catalyst bed. Some articles only determine the temperature of the external surface of the catalyst bed. Still, we intended to indicate the temperature gradient in the catalyst bed in both macro-scale (mm scale) and nanoscale.

It is difficult to real-time measurement of the inner temperature during MW in situ XAFS because the insertion of the fiber-optic thermometer in the catalyst bed interferes XAFS measurement. Therefore, we applied the combination of thermography and the simulation to obtain the inner temperature of the T_{in} . We have previously used this process to determine the overall temperature gradient in the catalyst bed (Figure 1, Haneishi et al., Ind. Eng. Chem. Res. 2017, 56, 27, 7685-7692, Figure 2, Haneishi et al., Sci. Rep. 2019, 9, 222). Other groups also use this simulation procedure for prediction of thermal run-away of zeolite (Nigar et al., Appl. Therm. Eng. 2019, 155, 5, 226-238) and local temperature of bubbles during liquid phase organic synthesis (De bruyn et al., J. Am. Chem. Soc. 2017, 139, 15, 5431-5436). The estimation of temperature by the coupled simulation nicely coincided with each other. Therefore, this is a useful method to estimate the overall temperature gradient in the catalyst bed.

Figure 1, (a) Simulated temperature distribution at a cross section of the catalyst bed when the core temperature was 600 °C. Simulated temperatures along (b) the vertical direction and (c) the horizontal direction. The red \bigcirc indicate measured temperatures corresponding to those in Figure 5. (Haneishi et al., Ind. Eng. Chem. Res. 2017, 56, 27, 7685-7692).

Figure 2, (a) Simulated electric field distributions in a cross-section of the microwave cavity including the C1-40 magnetite catalyst spheres, quartz tube, and antenna. Cross-sections of the simulated catalyst bed including the C1-40 magnetite catalyst spheres, void, and quartz tube showing the (b) electric field, (c) electromagnetic power loss density, and (d) temperature distributions. Input power: 5 W. (Haneishi et al., Sci. Rep. 2019, 9, 222)

As similar to Answer 4 for reviewer #2, T_{ex} was measured at the catalyst bed side not to interfere with the XAFS measurement (transmission mode). In addition, T_{ex} can only be measured during MW in situ EXAFS (gas-flow) and MW operando XANES in the present system. There is no way of inserting a fiber-optic thermometer probe in the catalyst pellet in the sealed gas-line for XAFS measurement. Moreover, the fiber-optic probe interferes with transmission XAFS measurement. Therefore, T_{ex} is an essential parameter for real-time monitoring of the temperature of the catalyst bed under MW heating at the same position in all experiments of in situ EXAFS and operando XANES.

Comment 3

In fact in the set up for conventional heating, different than the one for MW, it is not clear where they measured the temperature (see below): *"The in-situ XAFS measurement under CH was applied using an in-situ XAFS cell, which can heat a sample under gas flow (Supplementary Fig. 4a). The sample was inserted into a metal ring and was then placed inside the in-situ XAFS cell. The XAFS spectra were obtained in the 298–673 K temperature range, which was controlled by a thermocouple thermometer under air"* Thus, in both cases the authors are measuring similar average bulk temperature in the solid and the temperature difference of 100K that they claim in Figure 2e it is not correct.

Answer 3

We have revised the following explanation about the set-up for *in-situ* XAFS under CH regarding the temperature measurement.

"The *in-situ* XAFS measurement under CH was applied using an *in-situ* XAFS cell (KEK IMSS PF), which can heat a sample under gas flow (Supplementary Fig. 4a). The sample of 110 mg was pelletized in a metal ring and placed in the *in-situ* XAFS cell. A heat insulator surrounds the *in-situ* XAFS cell to maintain the homogeneous temperature. Two thermocouples measure the temperature at the sample and flowing gas to keep the same temperature during XAFS measurement. The XAFS spectra were obtained in the temperature range between 298–673 K."

Since the formation of the temperature gradient is critical for accurate comparison of T_{ex} , T_{in} , T_{Pt} , we have also added the details of T_{in} estimation by thermography and the simulation for as Figure 3 in the revised manuscript. We indicated both T_{Pt} and the T_{in} to compare the local and the bulk temperatures at the same part of the catalyst pellet for correct evaluation of the local heating of the Pt NPs under MW heating.

Reviewers' comments:

Reviewer #1 (Remarks to the Author):

The authors completely answers to all the questions doing additional experiments when possible and necessary.

Th paper can be accepted as it is.

Reviewer #2 (Remarks to the Author):

The paper was revised properly according to referees comments and the requested new information was added. I recommend to accept the manuscript in the present form.

Reviewer #3 (Remarks to the Author):

Comments in attatched file

Manuscript#: COMMSCHEM-20-0019-T

“Probing nano-temperature of supported Pt nanoparticles under microwave heating by in situ and operando XAFS”

Dear editor,

After reading carefully the revised version of the manuscript and the answers to the reviewers, I still strongly believe, based on the evidences presented by the authors that they are not measuring “the nano temperature of the supported Pt nanoparticles” as they claim in the paper instead they are measuring an average temperature of the Pt nanoparticles in a pellet with of a broad temperature distribution.

According to answer 1: “the current transmission XAFS is a bulk analysis with 1 mm × 1 mm of spot size and 4–6 mm thickness of the catalyst pellet. However, XAFS gives information on the average structure in the local region (up to around 6 Å), which means that the present MW in situ XAFS provides average structure local Pt NPs (nano-scale) in a bulk catalyst (millimeter scale).”

The authors confirm that the signal that they acquire gives information about **average structure** local Pt NPs (nano-scale), and this averaging information comes **from a millimeter region where there is a broad temperature distribution** according to its own simulations, now in Figure 3 e to g in the main text.

For this reason the Debye -Waller factor (σ^2) related to the average local temperature of the nanoparticles cannot be directly correlated neither to the Text (outside the pellet) or T_{in} (which is just a line in the central pellet).

The authors in answer 1 also justify their measurements based on previous results “Besides, similar work has been done by Van de Broek et al., who previously proposed that the Debye-Waller factor (σ^2) in EXAFS oscillations could be applied to estimate the temperate of the Au NPs suspension under Plasmonic heating (Small, 2011, 7, 2498-2506). This paper also indicates the use of the Debye-Waller factor from in situ EXAFS measurement enables estimation of the local temperature of Au NPs.”

Actually, I read the paper during my first revision and the conditions for the measurement in that paper definitely are not the same as in this case. In the work (Small, 2011, 7, 2498-2506) the measurement of the nanoparticles it is in a capillary tube containing 100 μ L of the nanoparticles suspended in water, in this capillary the temperature is HOMOGENEOUS and this is why they can correlated directly the Debye

-Waller factor (σ^2) of the nanoparticles to the temperature measured with a thermocouple. The system was calibrated initially and then when the nanostars are heated under illumination they observed a temperature difference of 2K on the temperature measured on the nanoparticles with respect to the aqueous solution. Definitely the system is not comparable at all.

The paper contains a lot of relevant information experimental and also simulations of temperature distribution with COMSOL. As a global it is a nice work but I do not agree on the conclusions extracted from the measurements performed. I totally agree with the authors about the difficulties in measuring real temperature in solids heated under MWs.

Finally, the paper could not be accepted in the revised version and only could be accepted if the authors reformulate conclusions, change title, and consider that they are not measuring the local temperature of the Pt nanoparticles, instead they are obtaining information of the average temperature of the Pt NPs in a bulk pellet.

Answers to the comments from Reviewer #1:

Comment 1

The authors completely answers to all the questions doing additional experiments when possible and necessary. The paper can be accepted as it is.

Answer 1

We are grateful to the positive reputation to our paper.

Answers to the comments from Reviewer #2:

Comment 1

The paper was revised properly according to referees comments and the requested new information was added. I recommend to accept the manuscript in the present form.

Answer 1

We are grateful to the positive reputation to our paper.

Answers to the comments from Reviewer #3:

Comment 1

After reading carefully the revised version of the manuscript and the answers to the reviewers, I still strongly believe, based on the evidences presented by the authors that they are not measuring “the nano temperature of the supported Pt nanoparticles” as they claim in the paper instead they are measuring an average temperature of the Pt nanoparticles in a pellet with of a broad temperature distribution.

According to answer 1: “the current transmission XAFS is a bulk analysis with 1 mm × 1 mm of spot size and 4–6 mm thickness of the catalyst pellet. However, XAFS gives information on the average structure in the local region (up to around 6 Å), which means that the present MW in situ XAFS provides average structure local Pt NPs (nanoscale) in a bulk catalyst (millimeter scale).”

The authors confirm that the signal that they acquire gives information about average structure local Pt NPs (nano-scale), and this averaging information comes from a millimeter region where there is a broad temperature distribution according to its own simulations, now in Figure 3 e to g in the main text.

For this reason the Debye -Waller factor (σ^2) related to the average local temperature of the nanoparticles cannot be directly correlated neither to the Text (outside the pellet) or Tin (which is

just a line in the central pellet).

Answer 1

Thank you very much for the comment. Our paper aims to discuss the local temperature by the Debye-Waller factor which is extracted from the local structural information determined by the in situ EXAFS. This is not the exact local temperature as pointed out by the Reviewer #3, however, this factor can be used to estimate the local temperature when the structure and oxidation state of Pt NP do not change. We clearly indicated the above discussion in the revised abstract and the main text, since the former version of our paper misleads that we are directly measuring the exact temperature of the supported PtNPs.

We have also modified “nano-temperature” in the title to “temperature” of the Pt NPs. We are selectively estimating the temperature of PtNPs, but not directly measuring the nano-temperature. Our former version was not well describing the difference between spectroscopic resolution and spatial resolution.

In addition, temperature gradient is formed in the catalyst bed under MWs. As indicated in Fig. 3g (Line 4), there are temperature gradient within 10 K at 453 K (T_{in}). Meanwhile, the temperature difference between the bulk (T_{in}) and PtNP (T_{Pt}) estimated from the DW factor reaches 26 K (Pt/ Al_2O_3) and 132 K (Pt/ SiO_2), respectively. This large temperature difference cannot be explained only by the temperature gradient in the pellet. We have also provided simulation of temperature gradient under CH in the Supplementary Fig. 18. In the isothermal condition, there was no temperature gradients observed under CH. So we can conclude that the significantly smaller DW factor are attributed to the local high temperature at the Pt NPs under MWs.

Yet, we are going to improve the MW in situ XAFS cell to minimize the temperature gradient by the insulators to obtain more precise local temperature. However, there are still difficulty in insulator that satisfies both transparencies to X-ray and MW at the same time.

Fig. 3 Determination of temperature distribution of Pt/Al₂O₃ catalyst pellet under MW heating. (a) Temperature mapping by the thermography under 18-watt MWs. (b) Schematic illustration of MW setup for measuring the temperature mapping of catalyst pellet by the thermography. (c) Reproduced temperature mapping by COMSOL Multiphysics simulation. (d) Temperature profiles of lines 1 and 2. (e) Simulated temperature mapping from a different angle. (f) Temperature profiles of lines 3 and 4. The average temperature of line 4 (T_{in}), where X-ray passed during *in-situ* EXAFS experiment.

Supplementary Fig. 18 Temperature distribution of Pt/Al₂O₃ under conventional heating.

Comment 2

The authors in answer 1 also justify their measurements based on previous results *“Besides, similar work has been done by Van de Broek et al., who previously proposed that the Debye-Waller factor (σ^2) in EXAFS oscillations could be applied to estimate the temperature of the Au NPs suspension under Plasmonic heating (Small, 2011, 7, 2498- 2506). This paper also indicates the use of the Debye-Waller factor from in situ EXAFS measurement enables estimation of the local temperature of Au NPs.”*

Actually, I read the paper during my first revision and the conditions for the measurement in that paper definitely are not the same as in this case. In the work (Small, 2011, 7, 2498-2506) the measurement of the nanoparticles it is in a capillary tube containing 100 μL of the nanoparticles suspended in water, in this capillary the temperature is HOMOGENEOUS and this is why they can correlated directly the Debye-Waller factor (σ^2) of the nanoparticles to the temperature measured with a thermocouple. The system was calibrated initially and then when the nanostars are heated under illumination they observed a temperature difference of 2K on the temperature measured on the nanoparticles with respect to the aqueous solution. Definitely the system is not comparable at all.

Answer 2

Yes, there is a difference between the previous report by Van de Broek et al. and ours however, the concept is the same. Van de Broek et al. applied the in situ EXAFS to the aqueous system so that the bulk temperature can be regarded as uniform.

On the other hand, our aim is to reveal the complexity of the temperature gradient formed in the packed bed of solid materials by MWs. Therefore, we have optimized the in situ XAFS system to be more suitable for MW heating of solid catalysts. We have combined several temperature monitoring systems (quartz transparent IR thermometer and microscope thermography) and Multiphysics simulation to carefully correct the temperature gradient in the pellet.

General XAFS provides only average information of any samples (nano-particles, capillary, powders...) as long as we do not use nano-scale incident light. (We are also interested in the nano-scale (beam size base) XAFS under MW to be more sensitive to the local Pt temperature.)

So, we discussed the formation of local temperature by changing the several parameters. The local high temperature of the Pt varied depending on the support (SiO_2 and Al_2O_3) and the thermal

conductivity of the atmosphere (N₂ and He). Moreover, the formation of local high temperature coincided with the reaction rate enhancement for dehydrogenation of 2-propanol. The faster reduction of Pt was also observed by MW operando XANES.

By summarizing the above results, we can conclude the MW irradiation forms of local high temperature of Pt NPs and contributes to the enhancement of the catalytic reactions.

Comment 3

The paper contains a lot of relevant information experimental and also simulations of temperature distribution with COMSOL. As a global it is a nice work but I do not agree on the conclusions extracted from the measurements performed. I totally agree with the authors about the difficulties in measuring real temperature in solids heated under MWs.

Answer 3

Thank you very much for the comment. Our challenge is to rationally understand the complex temperature gradient in the solid materials by MWs and reveal the mechanism of reaction rate enhancement. The formation of local high temperature has long been discussed in this field as described by the following representative papers;

Perry, W. L., Katz, J. D., Rees, D., Paffet, M. T., & Datye, A. K. Kinetics of the microwave-heated CO oxidation reaction over alumina-supported Pd and Pt catalysts. *J. Catal.* **171**, 431–438 (1997).

Zhang, X., Hayward, D. O., & Mingos, D. M. P. Apparent equilibrium shifts and hot-spot formation for catalytic reactions induced by microwave dielectric heating. *Chem. Commun.* **11**, 975–976 (1999).

Durka, T., van Gerven, T., & Stankiewicz, A. Microwaves in heterogeneous gas-phase catalysis: experimental and numerical approaches. *Chem. Eng. Technol.* **32**, 1301–1312 (2009).

Gangurde, L. S., Sturm, G. S. J., Devadiga, T. J., Stankiewicz, A. I., & Stefanidis, G. D. Complexity and challenges in noncontact high temperature measurements in microwave-assisted catalytic reactors. *Ind. Eng. Chem. Res.*, **56**, 13379–13391 (2017).

Our new challenge using MW in situ XAFS provides element-selective temperature information which cannot be done by other conventional thermometers.

Comment 4

Finally, the paper could not be accepted in the revised version and only could be accepted if the authors reformulate conclusions, change title, and consider that they are not measuring the local temperature of the Pt nanoparticles, instead they are obtaining information of the average temperature of the Pt NPs in a bulk pellet.

Answer 4

As similarly to the answers to the comments 1-3, we have modified the title, abstract and the discussion section in the revised version.

As pointed out by the reviewer, we are obtaining the average temperature of the Pt NPs in a bulk pellet. We clearly indicated that the temperature of Pt NPs were discussed from the DW factor which is dependent on the local temperature ($\sigma_{\text{temperature}}$) under particular condition when the structure of Pt do not change ($\sigma_{\text{structure}}$). The results clearly showed the formation of local high temperature which cannot be explained by the macro-scale temperature gradient in the pellet.

We have also changed several parameters and showed that the formation of local high temperature depends on the catalyst support and the atmosphere. We found that the formation of local high temperature coincides with the enhancement of catalytic dehydrogenation of 2-propanol by MWs.

By integrating the above results, we can conclude the formation of local high temperature at Pt NPs by MW heating. In the future work, we are going to improve the accuracy of the quantitative temperature assessment by XAFS.

REVIEWERS' COMMENTS:

Reviewer #3 (Remarks to the Author):

Editorial note: this reviewer provided no further comments for the authors.